# Recombinant Human Peptide Growth Factors, Bone Morphogenetic Protein-7 (rhBMP7), and Platelet-Derived Growth Factor-BB (rhPDGF-BB) for Osteoporosis Treatment in an Oophorectomized Rat Model

**DOI:** 10.3390/biom14030317

**Published:** 2024-03-07

**Authors:** Thamara Gonçalves Reis, Alice Marcela Sampaio Del Colletto, Luiz Augusto Santana Silva, Bruna Andrade Aguiar Koga, Mari Cleide Sogayar, Ana Claudia Oliveira Carreira

**Affiliations:** 1Cell and Molecular Therapy Group (NUCEL), School of Medicine, University of São Paulo, São Paulo 01246-903, SP, Brazil; thamara.gr23@gmail.com (T.G.R.); alicesampaiodelcolletto@gmail.com (A.M.S.D.C.); brunaaguiar5@gmail.com (B.A.A.K.); 2Biotechnology Graduate Program (PPG-USP), University of São Paulo, São Paulo 05508-900, SP, Brazil; 3Surgery Department, School of Veterinary Medicine and Animal Science, University of São Paulo, São Paulo 05508-270, SP, Brazil; 4PathoDxVet, São Paulo 02925-000, SP, Brazil; lzs1.augusto@gmail.com; 5Biochemistry Department, Chemistry Institute, University of São Paulo, São Paulo 05588-000, SP, Brazil

**Keywords:** osteoporosis, rhBMP-7, rhPDGF-BB, recombinant peptide growth factors, bone repair

## Abstract

Bone morphogenetic protein (BMP) and platelet-derived growth factor (PDGF) are known to regulate/stimulate osteogenesis, playing vital roles in bone homeostasis, rendering them strong candidates for osteoporosis treatment. We evaluated the effects of recombinant human BMP-7 (rhBMP7) and PDGF-BB (rhPDGF-BB) in an oophorectomy-induced osteoporosis rat model. Forty Sprague Dawley rats underwent oophorectomy surgery; treatments commenced on the 100th day post-surgery when all animals exhibited signs of osteoporosis. These peptide growth factors were administered intraocularly (iv) once or twice a week and the animals were monitored for a total of five weeks. Two weeks after the conclusion of the treatments, the animals were euthanized and tissues were collected for assessment of alkaline phosphatase, X-ray, micro-CT, and histology. The results indicate that the most promising treatments were 20 µg/kg rhPDGF-BB + 30 µg/kg rhBMP-7 twice a week and 30 µg/kg BMP-7 twice a week, showing significant increases of 15% (*p* < 0.05) and 13% (*p* < 0.05) in bone volume fraction and 21% (*p* < 0.05) and 23% (*p* < 0.05) in trabecular number, respectively. In conclusion, rhPDGF-BB and rhBMP-7 have demonstrated the ability to increase bone volume and density in this osteoporotic animal model, establishing them as potential candidates for osteoporosis treatment.

## 1. Introduction

Osteoporosis is a degenerative disease characterized by bone mass loss and tissue microarchitecture degeneration, leading to an increased risk of fractures, being considered a serious public health problem [1,2] that affects approximately 200 million people worldwide, with a higher prevalence in post-menopausal women and elderly men [3]. It is estimated that nearly one in three women and one in six men will experience an osteoporotic fracture in their lifetime. In the European Union alone, more than 23 million individuals are at high risk of fractures due to osteoporosis [4], resulting in over 8.9 million fractures globally each year [5]. Moreover, osteoporotic patients experience increased morbidity and mortality due to fractures and compromised bone tissue regeneration [6]. Notably, individuals who have suffered a recent fracture have a twofold higher probability of experiencing another one compared to their injury-free peers [7]. Furthermore, hip and vertebral fractures are associated with an elevated risk of death, with about 20% of individuals succumbing within six months of a hip fracture [8].

Bone is a unique organ in the body capable of regeneration and continual tissue remodeling throughout life [9,10]. Osteoclasts, responsible for resorbing bone; osteoblasts, involved in matrix secretion; and osteocytes, which ensure tissue homeostasis, are the pivotal cells in this process [11,12]. An imbalance between bone formation and resorption due to osteoblast and osteoclast activities, respectively, results in a loss of bone tissue mass and the deterioration of trabecular bone microarchitecture [13,14]. Thus, this imbalance results in osteoporosis due to incomplete bone pore filling from resorption, causing a progressive decrease in bone density [12].

Current osteoporosis treatments primarily fall into two categories: antiresorptive drugs that inhibit bone resorption and anabolic drugs that stimulate bone formation. Common medications include bisphosphonates (e.g., alendronate, risedronate, ibandronate, and zoledronic acid), denosumab, romosozumab, and teriparatide, in order of recommendation. While bisphosphonates are antiresorptive drugs, denosumab and romosozumab belong to the anabolic class [15]. However, these medications often come with adverse effects such as an increased risk of cardiovascular diseases [16], gastrointestinal issues [17], and osteonecrosis [18] derived from bisphosphonates; rebound effects characterized by bone volume reductions upon treatment discontinuation [19] due to denosumab use; and heart problems from romosozumab, which is contraindicated for patients with a history of myocardial infarction and stroke [20].

Since antiresorptive drugs do not promote bone tissue regeneration and typically require more than six months to show effectiveness [21], there is a critical need for the development of new anabolic treatments to facilitate bone mass recovery [9], particularly for patients unable to tolerate current medications. Our study investigates a class of molecules with potential against osteoporosis, namely, peptide growth factors, which play significant roles in bone regeneration and osteogenesis [22].

Bone morphogenetic proteins (BMPs) belong to the TGF-β superfamily and play essential roles in regulating bone and cartilage repair, embryonic development, bone tissue homeostasis [23,24], organogenesis, apoptosis, cell proliferation and differentiation, chemotaxis, tissue repair [25], and stem cell biology. These proteins are the most powerful and widely studied osteoinductive growth factors in regulating the formation of new bones [9]. Among them, BMP-7 induces the differentiation of mesenchymal stem cells into the osteoblastic lineage, leading to endochondral/intramembranous ossification and chondrogenesis [26]. It has also demonstrated efficacy in accelerating bone tissue regeneration and injury recovery in vitro, in vivo, and in preclinical studies [9,12,23].

Platelet-derived growth factor (PDGF) is a chemoattractant protein with mitogenic activity for mesodermal tissues, acting as a vascular anchorage agent and playing a vital role in bone and soft tissue wound healing [27]. It is also involved in central nervous system development, acting together with embryonic and developmental factors in promoting organogenesis [28]. PDGF-BB, in particular, plays a crucial role in fracture repair by promoting the infiltration of mesenchymal and angiogenic progenitor cells and regulating chondrogenic and osteogenic responses [29,30]. Additionally, PDGF-BB influences osteoblast chemotaxis, cell proliferation, and differentiation, facilitating rapid bone formation [27,30]. It is also associated with angiogenesis and osteogenesis [31], affecting bone shaping in the periosteum region and trabecular bone. Reduced PDGF-BB expression leads to the loss of trabecular and cortical bone mass [32].

Pountos et al. (2010) demonstrated that BMP-7 and PDGF-BB can increase mesenchymal stem cell proliferation and promote osteogenic differentiation when applied separately, fostering bone regeneration in in vitro models [33]. Therefore, we hypothesized that co-treatment with these two proteins could promote tissue recovery in an in vivo osteoporosis model, potentially leading to disease regression or attenuation.

Given the global prevalence of osteoporosis, its significant impact on patient morbidity and mortality, the challenges in repairing and regenerating osteoporotic fractures, and the associated healthcare costs, it is crucial to explore treatments capable of strengthening bones, preventing fractures, and promoting faster and more effective bone tissue repair. PDGF-BB and BMP-7 peptide growth factors have shown potential in regulating bone formation and tissue regeneration. Therefore, they are promising candidates for osteoporosis treatment, since they can stimulate bone formation, increase tissue density, and reduce the risk of fractures. Our primary objective was to assess the effects of rhBMP7 and rhPDGF-BB in an oophorectomy-induced osteoporosis rat model. Our results demonstrate the successful establishment of the osteoporosis animal model after 100 days of surgery. Additionally, systemic treatment with both recombinant proteins for five weeks (two weekly applications of 30 µg/kg of BMP-7 and 20 µg/kg of PDGF-BB) effectively increased bone volume and density in this osteoporotic model. To our knowledge, this is the first report showing that the combined systemic administration of these two recombinant growth factors partially reverses osteoporosis in this animal model, suggesting their potential as candidates for clinical treatment of this disease.

## 2. Materials and Methods

### 2.1. Growth Factors

The growth factors used in this study, specifically recombinant human platelet-derived growth factor (rhPDGF-BB) and recombinant human bone morphogenetic protein (rhBMP-7), were previously produced in our laboratory using mammalian cells (293T), following established protocols displaying high biological activity [34].

### 2.2. Animals and Treatment

For this investigation, we employed 44 female Sprague Dawley rats aged 12 weeks and weighing approximately 270 g each. These animals underwent a one-week acclimatization period and were maintained under constant environmental conditions, with a temperature of 22 ± 1 °C and a 12-h light/dark cycle. They had unrestricted access to both food and water.

The selection of the dosages for the peptide growth factors used in this study was based on a thorough review of the literature, considering studies that employed PDGF-BB and/or BMP-7 either intravenously or locally in the treatment of bone fractures in rat models [35,36,37,38,39,40,41].

Oophorectomy surgery was conducted according to established protocols found in the literature [42,43,44]. The study protocol was approved by the Ethics Committee on Animal Use (CEUA) at the School of Veterinary Medicine and Animal Science of the University of São Paulo (FMVZ-USP), under protocol number 8033260918. The animals were divided into eleven groups, as detailed in Table 1.

The surgical procedure was performed in the experimental animal facility using a horizontal laminar flow hood. The surgical area was meticulously disinfected with a solution of 70% ethanol and povidone iodine. Anesthesia was induced in the animals via inhalation of 2% isoflurane. Following a skin incision, access to the transverse abdominal muscle was gained and subsequent dissection of the muscle exposed the peritoneal space. After entry into the peritoneal cavity, both the uterus and ovaries were carefully exposed, and any excess adipose tissue surrounding these organs was removed. The uterus was ligated and both ovaries were excised. Once the uterus was repositioned within the peritoneal cavity, the surgical wound was closed and the internal planes were sutured. The rats’ weights were monitored weekly and blood samples were collected on days 0, 100, 115, 130, and 150.

After 100 days post-surgery, the animals in the oophorectomy group were euthanized through an intraperitoneal injection of an overdose of xylazine and ketamine to confirm the establishment of oophorectomy-induced osteoporosis. Treatments for the other groups commenced on the 100th day post-surgery. These treatments involved the injection of 30 µg/kg of rhBMP7, 20 µg/kg of rhPDGF-BB, or a combination of both, as detailed in Table 1.

The treatments were administered intravenously once or twice a week, depending on the designated experimental groups (Table 1). Each injection had a total volume of 150 µL and was delivered through the retro-orbital plexus of the animals. The solution used as the vehicle for all injections was PBSA (saline-phosphate solution without calcium or magnesium) and the treatment regimen was continued for a period of five weeks.

Two weeks after the completion of the treatment course, the animals were euthanized by an intraperitoneal injection of an overdose of xylazine and ketamine, marking a total of 150 days after osteoporosis induction surgery. Femurs were subsequently harvested and preserved in 10% formaldehyde for histological and micro-CT analysis.

### 2.3. Blood and Serum Collection

Gingival blood samples were collected from the animals on days 0 and 100 following oophorectomy surgery. The samples were placed in tubes containing separating gel, with a total volume of 500 µL. Approximately thirty minutes after collection, the blood samples were centrifuged at 1250× *g* for 10 min at 4 °C using an Eppendorf centrifuge-5810R (Eppendorf, Hamburg, Germany). The resulting serum was isolated, with approximately 250 µL collected and stored at −80 °C.

### 2.4. Quantification of Alkaline Phosphatase (ALP)

Alkaline phosphatase (ALP) serves as an enzyme linked to bone metabolism and can be used as an osteoporosis marker. ALP quantification was carried out employing the LabTest kit (LabTest, Lagoa Santa, MG, Brazil), following the manufacturer’s guidelines and adapting the protocol for the volume of solutions in 96-well plates. Spectrophotometric analysis (SpectraMax Paradigm, Molecular Devices, San Jose, CA USA) at a 590 nm wavelength was used to assess the results.

### 2.5. X-ray

On the 100th day following oophorectomy surgery, the animals underwent scanning using the In-Vivo Imaging System FX PRO (Bruker Corporation, Billerica, MA, USA) device. This generated X-ray images, which were subsequently analyzed by selecting the region of interest (ROI). The chosen region was the left femur neck, which corresponds to the region commonly examined for bone densitometry in humans. Variables such as bone volume content (g/cm^3^) were analyzed to quantify the bone mineral density (BMD) of the animals.

### 2.6. Micro CT

The left femurs were subjected to analysis using a microtomography device (SkyScan 1176 In Vivo Microtomograph, Bruker Corporation, Billerica, MA, USA). The obtained data were analyzed using the CT Analyzer software v. 1.18 (Blue Scientific, Cambridge, UK). The analysis focused on the proximal portion of the femoral epiphysis, involving 10 sections, with a combined thickness of about 10 µm. The ROI was used to select both the cancellous and cortical bone. The parameters quantified included bone mineral density (BMD), bone volume fraction (BV/TV; bone volume/total volume), trabecular thickness, and trabecular separation.

### 2.7. Histology

Tissues collected from the animals were washed in running water 48 h after collection to remove the fixative and subsequently stored in 70% ethanol. The right femur underwent decalcification, processing, paraffinization, cutting, and staining with hematoxylin and eosin (HE) and Masson’s Trichrome at the Histotech company in São Paulo, SP, Brazil.

Microscopic analysis and morphological diagnoses of the slides were conducted blindly. The microscopic findings were categorized using a scoring system as defined in Table 2. Additionally, the findings from each slide led to the suggestion of a clinical condition, which was also categorized according to a scoring system, as described in Table 3 [45,46,47].

### 2.8. Statistical Analysis

The collected data were subjected to statistical analysis employing GraphPad Prism v. 9 software (GraphPad Software, San Diego, CA, USA). Various statistical tests were employed, including the paired *t*-test, unpaired *t*-test, two-way ANOVA, and one-way ANOVA with Dunnett’s post-test, utilizing the Vehicle group as the reference. Significance was determined at a threshold of *p* < 0.05. Standard deviation (SD) was included in all presented graphs.

## 3. Results

### 3.1. Oophorectomy Induces Body Mass Gain

The oophorectomized animals exhibited significant and noteworthy weight gain 30 days after surgery (Figure 1). Treatment with recombinant proteins did not induce significant changes in the animals’ body mass, with no statistical differences observed between the groups (Figure 2).

### 3.2. Oophorectomized Animals Exhibit an Increased Serum ALP Concentration

A comparison of ALP concentration in animal serum at the oophorectomy day (D0) and 100 days after this procedure (D100) revealed a significant increase (*p* < 0.05) of 3 U/l, representing a 7% rise (Figure 3A). The mean ALP level at D0 was 44.38 (SD 11.05 and SEM 2.211), while at D100, it was 47.37 (SD 9.792 and SEM 1.958). This increase is expected and indicates alterations in bone metabolism in the animals.

### 3.3. In Vivo X-ray Analysis Confirmed the Osteoporosis Animal Model

The data from the in vivo X-ray analysis showed that 100 days after oophorectomy surgery, the animals experienced a significant loss of bone density compared to the animals in the Sham group (Figure 3B). Therefore, the animals were considered osteoporotic after 100 days post-surgery.

### 3.4. Micro-CT Analysis Indicates That Recombinant Protein Treatment Increases Bone Volume and Trabeculae Number

The results obtained from Micro-CT support the data from X-ray analysis, confirming that the animals became osteoporotic after 100 days of surgery. Oophorectomized animals had a significantly lower fraction of bone volume in the trabecular region compared to animals in the Sham group, with a 33% decrease in the analyzed region (Figure 3C). These values represent a 35% reduction in bone fraction relative to the Sham group.

When analyzing the cortical bone, oophorectomized animals exhibited a decrease in the bone volume fraction with a 12% reduction, corresponding to a 14% loss of bone compared to the Sham group (*p* < 0.01) (Figure 3D).

Upon evaluating the effects of different treatments, we found that the fraction of bone volume in the groups treated with BMP-7 2×/week and PDGF-BB + BMP-7 2×/week showed a significant increase, corresponding to 13% and 15%, respectively (Figure 4A). Treatment with BMP-7 2×/week and PDGF-BB + BMP-7 2×/week resulted in a significant increase (*p* < 0.05) in the number of trabeculae (Figure 4C), equivalent to 21% and 23%, respectively. When considering the BMD of treated animals relative to the Vehicle group, it can be noted that the treatments with PDGF-BB + BMP-7 2×/week and zoledronic acid showed a tendency to increase the animals’ BMD, although no statistical significance was found (Figure 4E).

### 3.5. Histology Evidences That Recombinant Proteins and Zoledronic Acid Can Reduce Osteoporosis Severity

Histological slides were prepared from the animals’ right femur. These slides were stained with HE, highlighting the cytoplasmic region and collagen fibers in pink/reddish, as well as the cell nuclei and extracellular matrix of the cartilage in purple. Additionally, we used Masson’s Trichrome stain, which indicated collagen in blue, cytoplasm in red, and cell nuclei in purple. In this analysis, the Control group served as our reference.

To confirm whether the animals were osteoporotic at 100 days post-oophorectomy, the Oophorectomy and Sham groups were compared to the Control (Figure 5A–F). The Sham group exhibited normal trabecular thickness (Figure 5G) and the number of trabeculae and medullary cellularity were within the expected range for the animals’ age, indicating normal bone tissue parameters and a normal clinical condition. Animals in the Oophorectomy group displayed a significant to moderate reduction in trabecular thickness compared to the Control group (Figure 5G), along with reductions in the number of trabeculae and medullary cellularity, as well as a considerable increase in intramedullary adipocytes, sometimes even showing a reduction in cortical thickness. These microscopic findings confirmed an osteoporotic condition, ranging from mild to severe (Figure 5H), confirming that 100 days post-oophorectomy, the animals indeed developed osteoporosis.

Additionally, in the histological analysis, all groups of oophorectomy surgery showed an increase in the number of intramedullary adipocytes and a reduction in medullary cellularity, which was more pronounced in animals with a greater reduction in trabecular bone thickness and more severe osteoporosis (Figure 6A–G and Figure 7A–G).

The animals in the Vehicle group displayed mild to moderate reduction in the articular cartilage (Figure 8A) and some animals also exhibited erosion (Figure 6A). Based on microscopic findings, the animals’ clinical condition indicated moderate osteoporosis and one case of severe osteoporosis (Figure 8B). The group treated with zoledronic acid, the positive control in this study, had one animal with a pronounced reduction in trabecular thickness, one with discrete reduction, and two with normal trabecular thickness (Figure 8A). The clinical condition followed the pattern of thickness reduction, with one animal presenting severe osteoporosis, one with mild osteoporosis, and two animals with a normal clinical condition (Figure 8B).

Two animals in the group that received PDGF-BB 2×/week had a pronounced reduction in trabecular thickness, one with moderate reduction and one with normal thickness of the trabeculae (Figure 8A). Analysis of the clinical condition of these animals indicated that both displayed severe osteoporosis, one had a moderate condition of the disease and the other was considered normal (Figure 8B). There were signs of trabecular fractures, cartilage erosion, and a considerable increase in the number of adipocytes throughout the animals of this group. The group that received PDGF-BB 1×/week exhibited a moderate reduction in the thickness of bone trabeculae in two animals and the other two presented a discrete reduction in trabecular tissue (Figure 8A). Regarding their clinical condition, two of these animals displayed moderate osteoporosis, another had mild osteoporosis, and one animal had a normal clinical condition (Figure 8B). The histological slides showed occasional cementation lines, which are the boundaries between the old and new bone matrix, neoformation of bone tissue, apparently intact cortical bone, and a focus on clustered regenerative chondrocytes, in addition to a considerable increase in intramedullary adipocytes (Figure 6D).

Treatment with BMP-7 2×/week resulted in varied reductions in trabecular thickness, with each animal presenting a different score (Figure 8A). Areas of neoformed trabecular bone tissue and cementing lines were observed. Regarding the clinical condition, two animals were considered normal, one had moderate osteoporosis, and the other displayed severe osteoporosis (Figure 8B). When treated with BMP-7 1×/week, the animals showed a moderate reduction in trabecular thickness and one animal had a discrete reduction (Figure 8A). This group also presented regions of newly formed bone with young osteocytes and an increase in osteoblasts and osteoclasts, even in animals with a moderate reduction. However, one of the animals presented fragmentation of the bone trabeculae. Upon observing the results (Figure 8B), it may be noted that one of the animals in this group had severe osteoporosis, two animals displayed moderate osteoporosis, and one had a clinical condition of mild osteoporosis (Figure 8B).

By associating the two recombinant proteins, namely PDGF-BB and BMP-7 2×/week, in the treatment of the animals, one animal displayed a pronounced reduction in trabecular thickness, another had a moderate reduction, and the last two exhibited only a discrete reduction (Figure 8A). Histological analysis showed that the group receiving PDGF-BB and BMP-7 1×/week had the best performance among those treated with recombinant proteins, with three animals classified as having a mild reduction in trabecular thickness, and one with a moderate reduction (Figure 8A). The presence of cementation lines, areas of bone neoformation, an increased number of adipocytes, and erosion in the articular cartilage were also observed.

## 4. Discussion

Osteoporosis is a disease characterized by the loss of bone mass and tissue microarchitecture, rendering it more fragile and prone to fractures. It is difficult to repair due to changes in bone metabolism and primarily affects menopausal women and men over 70 years of age. This condition arises from an imbalance between bone formation and resorption, leading to the formation of pores in the tissue, reducing its resistance [48].

Among the available therapies, bisphosphonates are anti-resorptive drugs that act on osteoclasts, inhibiting their activity, and are more widely used. However, these drugs only stabilize the loss of bone tissue and are incapable of inducing the replacement of lost bone mass [49]. In the pursuit of new therapies capable of increasing tissue volume and density, growth factors such as BMP-7 and PDGF-BB have shown promise. These factors play roles in the proliferation and differentiation of osteoblasts, contributing to tissue formation through secretion of bone matrix. Therefore, our study aimed to investigate the effects of recombinant human proteins PDGF-BB and BMP-7 on the progression of osteoporosis in oophorectomized animals.

After performing the oophorectomy surgery, the animals exhibited a significant and marked increase in weight, particularly 30 days after the procedure (Figure 1). This characteristic response of oophorectomized rats, as described in the literature, confirms the success of the surgery and the establishment of the disease [50,51,52]. After 90 days post-surgery, the animals’ weight reached a plateau, with minimal variation thereafter, observed in both oophorectomized animals and surgical control animals (Sham).

On the 100th day post-surgery, the serum ALP levels showed a significant increase, as expected with the onset of the disease, indicating a disruption in bone metabolism [53,54,55,56]. The predominant inorganic component in the bone matrix is hydroxyapatite, whose synthesis is inhibited by pyrophosphate. ALP is synthesized by osteoblasts and acts by hydrolyzing pyrophosphate, releasing inorganic phosphate that participates in hydroxyapatite synthesis [57]. In osteoporosis, where an imbalance between tissue formation and resorption occurs, an increase in osteoblast-mediated bone formation is expected and the concentration of ALP reflects the activity of these cells [54]. Furthermore, the total quantification of enzyme activity can serve as a complementary test for diagnosing osteoporosis.

In addition to the changes in ALP levels, we observed a reduction in bone volume and density through in vivo X-ray (Figure 3B) and ex vivo Micro-CT (Figure 3C,D) analyses of the left femur. These findings indicate that the animals had developed osteoporosis, successfully establishing the model at the designated time point.

This animal model closely mimics post-menopausal osteoporosis, in which, as in women, the animals experience a drastic reduction in estrogen concentration. Estrogen plays a crucial role in bone metabolism, associated with increased expression of RANKL by B lymphocytes. RANKL is a transcription factor that promotes the differentiation of osteoclasts, leading to increased bone resorption. Low estrogen levels are also linked to reduced expression of osteoprotegerin (OPG), a key RANKL inhibitor [58,59], further exacerbating the imbalance between bone resorption and formation.

Among the molecules with the potential to treat the disease are the peptide growth factors PDGF-BB and BMP-7. PDGF-BB is a chemotactic and mitogenic growth factor that promotes cell proliferation and migration of mesenchymal cells such as fibroblasts and mesenchymal stem cells (MSCs) [28]. Pre-osteoclasts secrete PDGF-BB during their differentiation process [60,61], inducing MSC migration and differentiation into osteoblasts. This process stimulates cortical bone formation during tissue damage and the formation of type H vessels, which are crucial for bone remodeling [32]. Studies have also indicated that the use of PDGF-BB increases osteoclast differentiation, both in vitro and in vivo. PDGF-BB acts as a chemotactic agent for osteoclast precursors [62], enhancing osteoclast resorption activity through the PDGF receptor beta (PDGFR-β) in vitro [63]. Therefore, PDGF-BB plays a multifaceted role in bone remodeling.

The application of PDGF-BB at 20 µg/kg 1×/week showed more significant improvements in trabecular tissue compared to the 2×/week treatment across all regions. Given the functions performed by this factor, once weekly applications can stimulate bone remodeling and osteoclast proliferation. However, with twice-weekly treatment, PDGF-BB may become a more significant stimulus for osteoclast differentiation, which acts more rapidly than osteoblasts on bone tissue, potentially leading to an imbalance in bone metabolism characteristic of osteoporosis.

BMPs are essential for bone metabolism and repair [64], exerting mitogenic, chemotactic, and proliferative activities, with the ability to induce MSC differentiation into osteoblasts. BMP-7, a member of the BMP family, has demonstrated remarkable potential in bone fracture repair [65,66,67] and can induce osteoblast differentiation even more effectively than PTH hormone [33], a widely used therapeutic alternative for osteoporosis treatment.

Considering the role of BMP-7 in osteoblasts, its use in treatments is expected to benefit osteoporotic animals. Among the BMP-7 treatment regimens tested in this study, the most effective was BMP-7 administered at 30 µg/kg twice a week, resulting in a 15% increase in bone volume fraction in the second region. Therefore, a weekly application of the recombinant protein proved insufficient to improve bone tissue quality.

The combination of PDGF-BB (20 µg/kg) and BMP-7 (30 µg/kg) was expected to increase bone volume and density, as PDGF-BB has mitogenic capacity and BMP-7 acts in MSCs differentiation into osteoblasts. When administered twice weekly, this treatment produced positive results in the first region analyzed by micro-CT. Moreover, when applied only once a week, significant improvements were observed in all regions, highlighting the significance of treatment frequency in inducing bone volume increases and achieving positive clinical outcomes.

These results also suggest the possibility of antagonistic interactions between PDGF-BB and BMP-7 when administered simultaneously, potentially hindering their functions. Chan et al. (2010) demonstrated that PDGF-BB may act as an antagonist of BMPs and the TGF-β family by reducing the expression of Trb3 (Tribbles-like protein-3), a modulator of Smad protein expression, leading to a decrease in the expression of these proteins and a subsequent reduction in BMP activity [68].

Bayer et al. (2016) highlighted the importance of the order of the administration of PDGF-BB and BMP-2, which belong to the same family as BMP-7, in in vitro angiogenic repair, indicating that specific administration regimens are crucial for achieving favorable outcomes. They also observed antagonistic effects between the signaling of these two proteins when administered simultaneously [69]. Another study by Novak et al. (2023) showed that the combination of these two growth factors compromises the bone healing model, with PDGF-BB inhibiting BMP-2-induced osteogenesis [70]. This effect could be associated with PDGF-BB’s inhibition of the BMP2/Smad canonical signaling pathway [61].

The histological analysis results indicated a significant increase in intramedullary adipocytes and a reduction in cellularity in the bone marrow in all groups of oophorectomized animals. MSCs within the bone marrow can differentiate, mainly into osteoblasts, adipocytes, and chondrocytes [71]. This differentiation is finely regulated and mediated by several factors and signaling pathways. Transcription factors include Runx2, which induces MSC differentiation into osteoblasts while inhibiting adipogenesis [72] and PPARγ, which promotes the reverse process by inducing differentiation into adipocytes and inhibiting osteoblastogenesis [73]. BMPs and the Wnt/β-catenin signaling pathways also play essential roles in this process, inducing MSC differentiation into both cell types. It has been observed that increased BMP-2 concentration induces osteoblastogenesis, while low concentrations promote adipogenesis [74], both through the canonical Smad 1/5/8 and MAPK signaling pathways. BMPs also promote adipogenesis by activating PPARγ [75]. The Wnt pathway induces Runx2 expression and inhibits PPAR expression, promoting osteoblast differentiation; however, when suppressed, it inhibits osteoblastogenesis and promotes the differentiation of MSCs into adipocytes. Other factors that regulate cell differentiation include miRNAs, high-fat diets, and mechanical stimulation [76]. In this context, studies have shown that MSCs from osteoporotic patients have an increased potential to differentiate into adipocytes at the expense of osteoblasts [77]. Other studies have shown that these adipocytes are distinct from those found elsewhere in the body and express RANKL, which is a significant factor in osteoclast differentiation and activity, as well as bone resorption [78].

The study by Hu and colleagues made important discoveries about how RANKL expressed by intramedullary adipocytes influences bone remodeling. Using a mouse model of oophorectomized females with RANKL deletion in adipocytes, they found that these animals did not exhibit the reduction in trabecular and cortical bone density observed in oophorectomized control animals. They concluded that RANKL produced by adipocytes plays an essential role in bone resorption and the increased differentiation of osteoclasts in bone tissue, thus contributing significantly to the development of osteoporosis [79]. Additionally, in a genetically mutated animal model, Yu and colleagues observed that, in addition to reducing bone resorption and osteoclastogenesis, the deletion of RANKL synthesis in these cells did not affect MSC differentiation into adipocytes or osteoblasts [80]. Therefore, this fact supports the link between increased intramedullary adipogenesis in osteoporotic animals and an imbalance in bone remodeling, leading to osteoporosis.

Furthermore, the histological data also indicate that the most effective treatments for osteoporosis were the administration of zoledronic acid and PDGF-BB and BMP-7 1×/week. Like the micro-CT results, the group treated with the commercial drug showed significant improvements, even enhancing the quality of the animals’ femurs. Comparable results were obtained by Black et al. (2007), who observed an increase in BMD in patients using zoledronic acid annually for three years, with these effects being observed in various areas, including the femur and hip [81] Other studies have also demonstrated the positive effects of zoledronic acid on bones, increasing BMD [82].

In conclusion, the induction of osteoporosis through oophorectomy was successful, as evidenced by increased animal weight, elevated blood ALP concentration, decreased BMD and bone volume fraction, and histological analysis indicating reduced trabecular thickness, increased intramedullary adipocytes, and a clinical condition compatible with osteoporosis. Regarding the investigated treatments, they did not cause changes in animal weight and micro-CT data revealed that BMP-7 2×/week and PDGF-BB + BMP-7 2×/week led to an increase in bone volume fraction and trabecular number. Finally, histomorphology indicated that the most effective treatments also include zoledronic acid and PDGF-BB + BMP-7 1×/week.

## 5. Conclusions

An animal model for osteoporosis was successfully established. On the 100th day after the oophorectomy surgery, the animals displayed a reduction in bone density and volume, along with an increase in the concentration of ALP, a high number of intramedullary adipocytes, and a significant reduction in trabeculae, indicating an osteoporotic state. Systemic treatment with recombinant proteins for five weeks, involving two weekly applications of 30 µg/kg of BMP-7 and 20 µg/kg of PDGF-BB, was effective in partially reversing osteoporosis. This treatment increased the fraction of bone volume by 15% and trabeculae number by 23%. To the best of our knowledge, this is the first study to employ these recombinant proteins concomitantly and systemically for osteoporosis treatment. We anticipate that this research will contribute to the development of a new and effective clinical treatment regimen, ultimately enhancing the quality of life for individuals affected by osteoporosis.

## Figures and Tables

**Figure 1 biomolecules-14-00317-f001:**
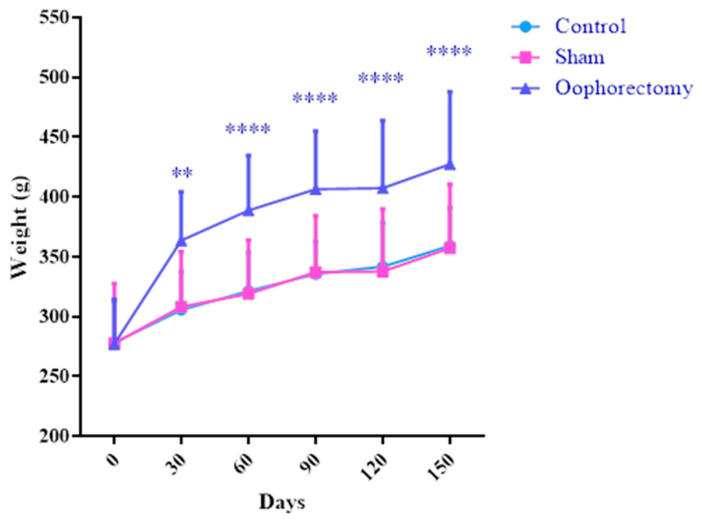
Comparison of weight gain on day 0 between oophorectomized animals, the control groups, and the Sham group. Two-way ANOVA test, **: *p* < 0.01; ****: *p* < 0.001. Control: N.:4; Sham: N.:4;. Oophorectomy: N.:36. N.: number of animals/group.

**Figure 2 biomolecules-14-00317-f002:**
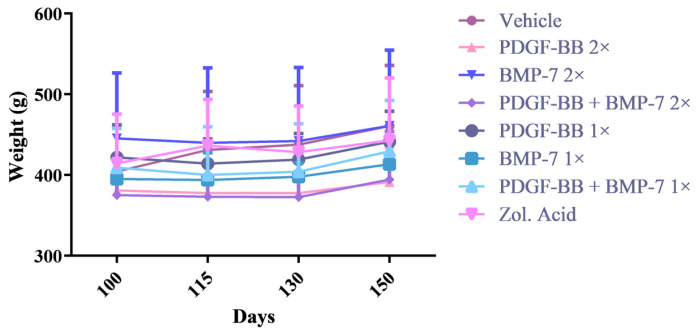
Comparison of weight gain during treatment among the different treatment groups. Two-way ANOVA test. N.: 4. Zol. Ac.: Zoledronic Acid.

**Figure 3 biomolecules-14-00317-f003:**
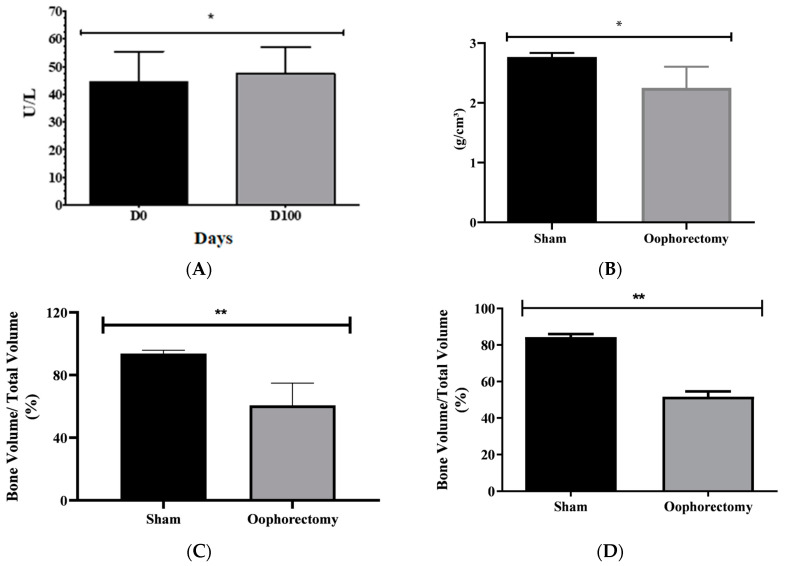
Oophorectomy induces osteoporosis after 100 days of surgery. (**A**): Alkaline phosphatase (ALP) concentration in animal serum on days 0 (D0) and 100 (D100) after oophorectomy surgery. U/L= Units per liter. Paired *t*-test, *: *p* < 0.05. N.: 32. (**B**): X-Ray analysis. On the 100th day after surgery (D100), a significant reduction in the bone mineral density of the femoral neck was observed in the oophorectomized animals, when compared to the Sham group. Unpaired *t*-test, *: *p* < 0.05. N. Oophorectomy: 32, N. Sham: 4. (**C**): Comparison of the bone volume fraction (BV/TV) of the trabecular region between animals in the Sham and Oophorectomy group, expressed as a percentage. Unpaired *t*-test, **: *p* < 0.01. N.: 4. (**D**): Comparison of the bone volume fraction (BV/TV) of the cortical region between the animals in the Sham and Oophorectomy groups, expressed as a percentage. Unpaired *t*-test, **: *p* < 0.01. N.: 4. N.: number of animals/group.

**Figure 4 biomolecules-14-00317-f004:**
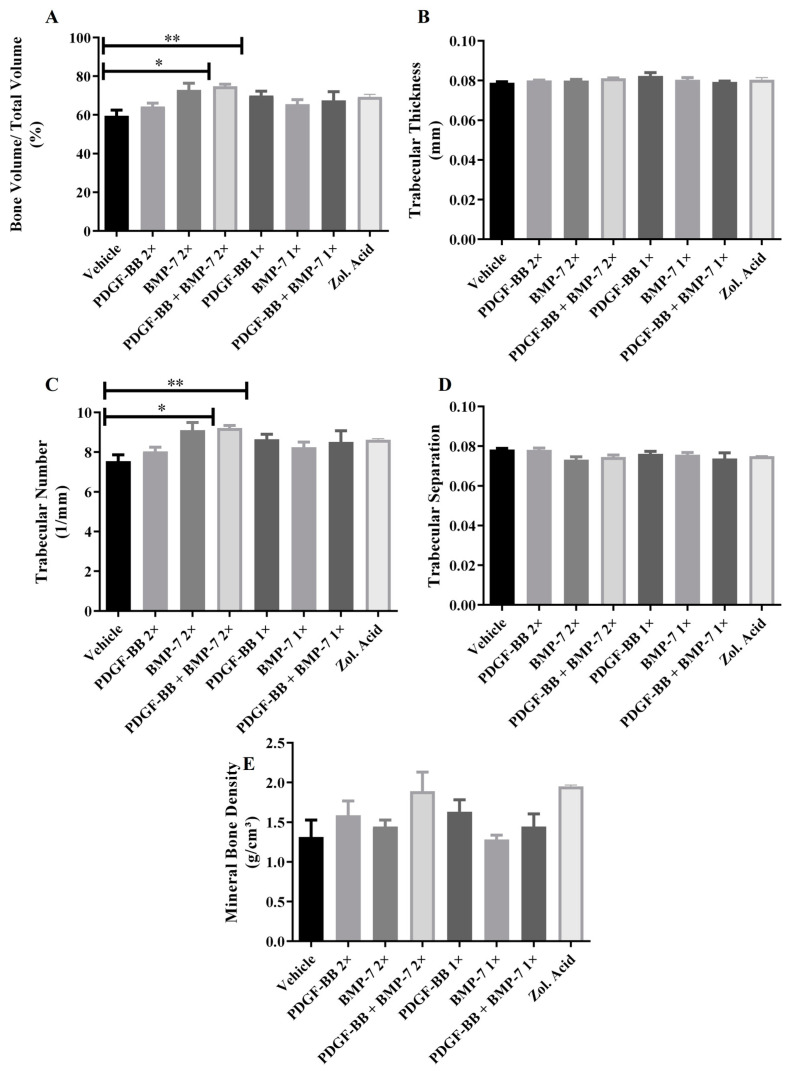
Micro-CT analysis of the proximal portion of the femoral epiphysis (first Region). (**A**): Bone volume fraction; (**B**): Thickness of trabeculae; (**C**): Number of trabeculae; (**D**): Separation of trabeculae; (**E**): Bone mineral density (BMD). One-way ANOVA Test and Dunnett’s Test against Vehicle, *: *p* < 0.05, **: *p* < 0.01. N = 4. Zol. Acid: Zoledronic acid. N.: number of animals/group.

**Figure 5 biomolecules-14-00317-f005:**
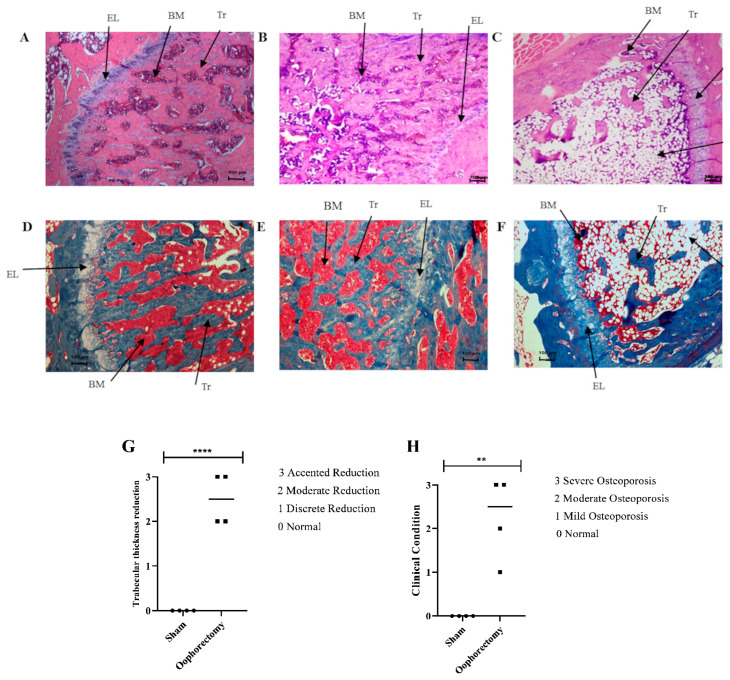
Histological data indicating the development of osteoporosis in oophorectomized animals. (**A**): Control stained with HE; (**B**): Sham stained with HE; (**C**): Oophorectomy stained with HE; (**D**): Control stained with Masson’s Trichrome; (**E**): Sham stained with Masson’s Trichrome; (**F**): Oophorectomy stained with Masson’s Trichrome; (**G**): Score of the clinical status of controls referring to histopathological analysis; **: *p* < 0.01, ****: *p* < 0.001 (**H**): Controls trabecular reduction score for histopathological analysis. N.: 4. Scale bar: 100 µm. Ad: Adipocytes; BM: Bone marrow; Tr: Trabeculae; EL: Epiphyseal line; N.: number of animals/group; Circle: Sham group animal; Square: Oophorectomy animal group; Line: Group median. Each square represents one animal, the double square symbol corresponds to two animals.

**Figure 6 biomolecules-14-00317-f006:**
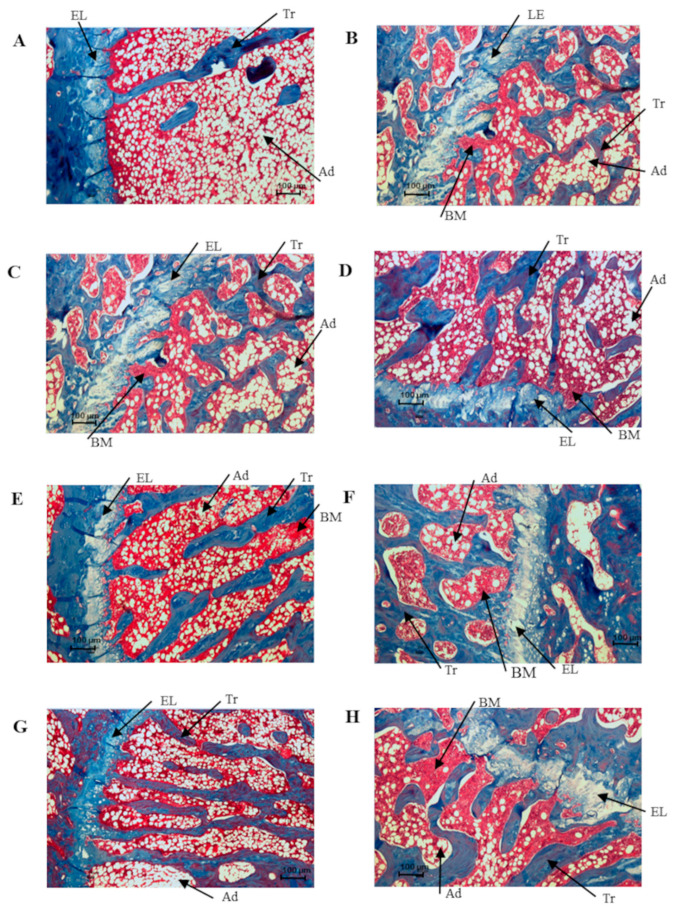
Masson’s trichrome stained histological slides comparing the different treatments. (**A**): Vehicle; (**B**): Zoledronic acid; (**C**): PDGF-BB 2×/week; (**D**): PDGF-BB 1×/week; (**E**): BMP-7 2×/week (**F**): BMP-7 1×/week; (**G**): PDGF-BB + BMP-7 2×/week (**H**): PDGF-BB + BMP-7 1×/week. Scale bar: 100 µm. In blue are collagen fibers, cytoplasm in red, and cell nuclei in purple. Ad: Adipocytes; BM: Bone marrow; Tr: Trabeculae; EL: Epiphyseal line.

**Figure 7 biomolecules-14-00317-f007:**
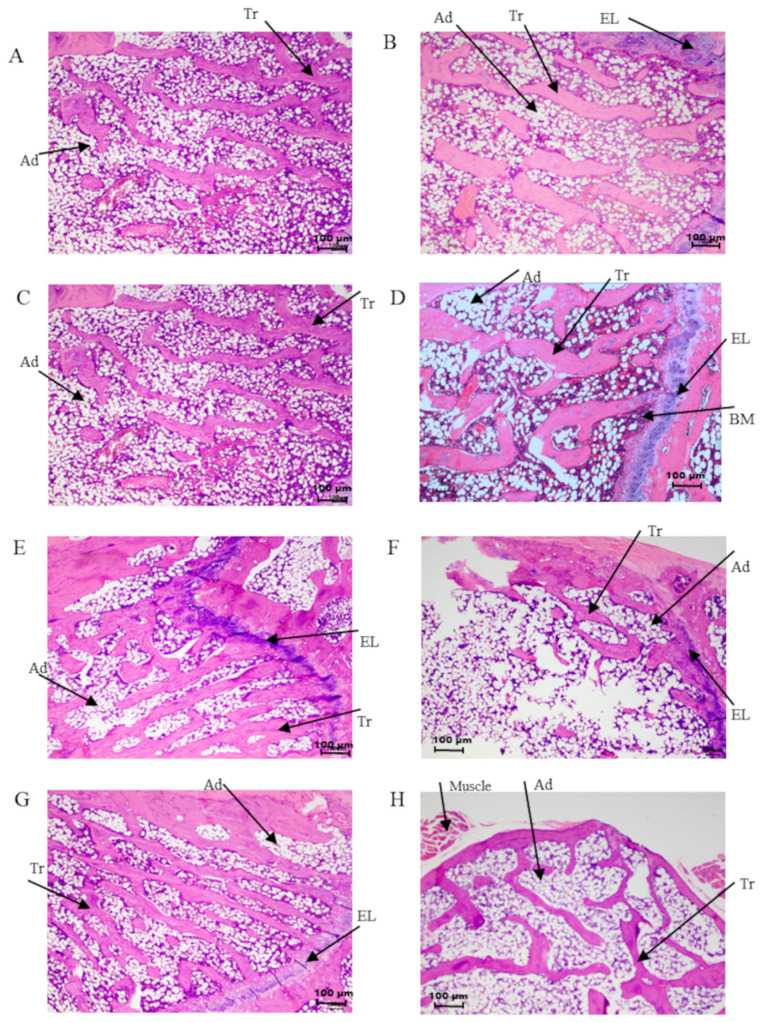
Histological slides stained with HE comparing the different groups analyzed. (**A**): Vehicle; (**B**): Zoledronic acid; (**C**): PDGF-BB 2×/week; (**D**): PDGF-BB 1×/week; (**E**): BMP-7 2×/week (**F**): BMP-7 1×/week; (**G**): PDGF-BB + BMP-7 2×/week (**H**): PDGF-BB + BMP-7 1×/week. Scale bar: 100 µm. In pink/reddish are the cytoplasm and collagen fibers, in purple were stained the cell nuclei. Ad: Adipocytes; BM: Bone marrow; Tr: Trabeculae; EL: Epiphyseal line.

**Figure 8 biomolecules-14-00317-f008:**
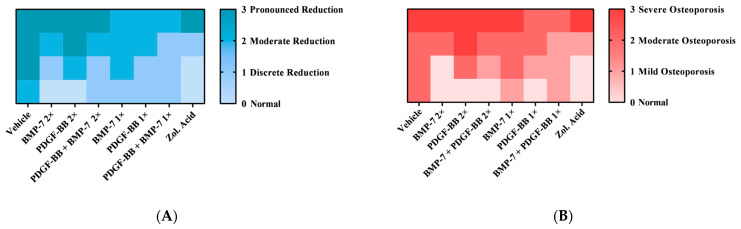
Treatment with recombinant proteins led to increased trabecular thickness and reduced disease severity. (**A**): Trabecular thickness scores of different treatment groups referring to histomorphological analysis. (**B**): Scores of the clinical status of the different treatment groups regarding the histopathological analysis.

**Table 1 biomolecules-14-00317-t001:** Treatment group division.

Group	Surgery	Treatment	Number of Injections	Dose/Injection	Euthanasia
Control	Not performed	None	Non-applicable	Non-applicable	150 days
Sham	Surgery without ovary remotion	None	Non-applicable	Non-applicable	150 days
Zoledronic Acid	Oophorectomy	Zoledronic acid	Two injections within a 22-day interval	100 µg/kg	150 days
Vehicle	Oophorectomy	None	2×/week	20 mM Tris-HCl pH 7.2 + 300 mM NaCl	150 days
PDGF-BB 2×	Oophorectomy	PDGF-BB	2×/week	20 µg/kg	150 days
BMP-7 2×	Oophorectomy	BMP-7	2×/week	30 µg/kg	150 days
PDGF-BB + BMP-7 2×	Oophorectomy	PDGF-BB + BMP-7	2×/week	20 µg/kg of PDGF-BB + 30 µg/kg of BMP-7	150 days
PDGF-BB 1x	Oophorectomy	PDGF-BB	1×/week	20 µg/kg	150 days
BMP-7 2×	Oophorectomy	BMP-7	1×/week	30 µg/kg	150 days
PDGF-BB + BMP-7 1×	Oophorectomy	PDGF-BB + BMP-7	1×/week	20 µg/kg of PDGF-BB + 30 µg/kg of BMP-7	150 days
Oophorectomy	Oophorectomy	None	Non-applicable	Non-applicable	100 days

**Table 2 biomolecules-14-00317-t002:** Scores of trabecular thicknesses observed in the histomorphological analysis.

Score	Trabecular Thickness
0	Normal
1	Discrete reduction
2	Moderate reduction
3	Pronounced reduction

**Table 3 biomolecules-14-00317-t003:** Clinical conditions derived from the microscopic findings and their respective scores.

Score	Clinical Condition
0	Normal
1	Mild osteoporosis
2	Moderate osteoporosis
3	Severe osteoporosis

## Data Availability

Data can be made available upon request from the corresponding author.

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
