# Peer review of "Recombinant Human Peptide Growth Factors, Bone Morphogenetic Protein-7 (rhBMP7), and Platelet-Derived Growth Factor-BB (rhPDGF-BB) for Osteoporosis Treatment in an Oophorectomized Rat Model"

_biomolecules, 2024, doi:10.3390/biom14030317_

Round 1

Reviewer 1 Report

Comments and Suggestions for Authors

This research investigated the impact of recombinant human BMP-7 (rhBMP7) and PDGF-BB (rhPDGF-BB) on osteoporosis using a rat model. The study involved forty Sprague-Dawley rats that were surgically induced with osteoporosis. Treatment began 100 days after surgery. Results indicate that rhPDGF-BB and rhBMP-7 significantly improve bone volume and density in osteoporotic rats, highlighting their potential as effective treatments for osteoporosis.

Critical observations regarding the study include:

  • The introduction needs a distinct statement of its novel contribution to the field.
  • The study lacks sufficient references to related research. Expanding the introduction with recent studies and a comparative analysis of current osteoporosis treatments would be beneficial, emphasizing the importance of the selected treatment methods.
  • The rationale behind the specific concentrations of BMP-7 and PDGF-BB used should be clarified in the methods or introduction.
  • The abbreviation 'HE' used in line 170 requires definition.
  • The methods section is missing details on how statistical calculations were performed.
  • Figure 3 could be misleading as significant differences are not clear. Including numerical data with error margins in the text would be helpful.
  • It's recommended to combine Figures 3, 4, 5, and 6 for better readability and flow.
  • Figures 9 A-G and 10 A-G each need distinct explanations.
  • Merging Figures 11 and 12 could improve clarity.
  • The section from lines 428-465 seems unrelated to the study's focus on RANKL expression or cell differentiation and could be considered irrelevant.
  • The discussion references are outdated. It's important to compare findings with more recent studies.
  • The conclusion should be revised to include key results, numbers, and the study's novelty for a stronger impact.

Author Response

Please see the document attached.

Reviewer 2 Report

Comments and Suggestions for Authors

This original article aimed to assess the effects of recombinant human BMP-7 and PDGF-BB in oophorectomy-induced osteoporosis in rats. The authors concluded that these factors have the capability to reverse osteoporosis in a rat model. While the paper is somewhat interesting and could be considered for publication in Biomolecule, several minor improvements should be made based on the following comments.

Minor comments:

  1. Please revise the abstract, with particular attention to clarifying the rationale of the current study.
  2. Please clearly explain what are the novel results obtained in your present study in the manuscript.
  3. Provide the full name of each abbreviation upon its first appearance in the main text.
  4. Please provide concise and understandable legends for each figure.
  5. Please insert a scale bar in each picture used in the figures.

Comments on the Quality of English Language

The manuscript should be checked by the native English speakers for re-submission.

Author Response

Please see the document attached.
